# A Primary Kidney Giant Cell Tumor of Soft Tissue Caused Peritoneal Dissemination, Considered to Be Malignant Transformation: A Case Report

**DOI:** 10.3390/diagnostics13040752

**Published:** 2023-02-16

**Authors:** Chiina Hata, Yuki Fukawa, Toru Motoi, Yuko Kinowaki, Takumi Akashi, Kenichi Ohashi, Yudai Ishikawa, Yuma Waseda, Yasuhisa Fujii, Ryota Kakuta, Sadakatsu Ikeda, Iichiroh Onishi

**Affiliations:** 1Department of Human Pathology, Graduate School of Medical and Dental Sciences, Tokyo Medical and Dental University, Tokyo 113-8510, Japan; 2Department of Oral Pathology, Graduate School of Medical and Dental Sciences, Tokyo Medical and Dental University, Tokyo 113-8510, Japan; 3Department of Pathology, Tokyo Metropolitan Cancer and Infectious Disease Center, Komagome Hospital, Tokyo 113-8677, Japan; 4Department of Comprehensive Pathology, Graduate School of Medical and Dental Sciences, Tokyo Medical and Dental University, Tokyo 113-8510, Japan; 5Department of Diagnostic Pathology, Tokyo Medical and Dental University Hospital, Tokyo 113-8510, Japan; 6Department of Urology, Tokyo Medical and Dental University Hospital, Tokyo 113-8510, Japan; 7Department of Clinical Oncology, Graduate School of Medical and Dental Sciences, Tokyo Medical and Dental University, Tokyo 113-8510, Japan

**Keywords:** giant cell tumor of soft tissue, kidney, malignant transformation

## Abstract

Giant cell tumor of soft tissue (GCTST) is a defined disease entity that has a morphology similar to giant cell tumor of bone (GCTB). The malignant transformation of GCTST has not been reported, and a kidney primary is extremely rare. We report the case of a 77-year-old Japanese male, who was diagnosed with primary GCTST of the kidney and showed peritoneal dissemination, considered to be a malignant transformation of GCTST, in 4 years and 5 months. Histologically, the primary lesion showed characteristics of round cells with not prominent atypia, multi-nucleated giant cells, and osteoid formation, and carcinoma components were not found. The peritoneal lesion was characterized by osteoid formation and round to spindle-shaped cells, but differed in nuclear atypia, and multi-nucleated giant cells were not detected. Immunohistochemical and cancer genome sequence analysis suggested these tumors were sequential. This is a first report of a case that we could diagnose as primary GCTST of the kidney and could be determined as malignant transformation of GCTST in the clinical course. Analysis of this case will be examined in the future when genetic mutations and the disease concepts of GCTST are established.

## 1. Introduction

Giant cell tumor of soft tissue (GCTST) is a very rare tumor with a predilection for the subcutaneous areas of the extremities. The histology resembles that of giant cell tumor of bone (GCTB), but it is thought to be genetically distinct from GCTB, because the *H3F3A* mutation is not present. Histologically, it is composed of round mononuclear cells and multinucleated osteoclast-like giant cells, and about half of the patients show bone formation [1]. Malignant transformation is very rare and difficult to determine.

Tumor with giant cells in the kidney needs to be distinguished from giant cell tumor and giant cell urothelial carcinoma. In this report, we describe a case of primary giant cell tumor of the kidney, which showed postoperative peritoneal dissemination, and was considered to be a malignant transformation.

## 2. Case Presentation

### 2.1. Clinical Summary

A 77-year-old man was found to have a left renal pelvis mass (6.5 cm × 5.0 cm × 4.0 cm) on contrast-enhanced CT for close examination of a gallbladder polyp. He had undergone a left nephrolithotomy at age 53 and extracorporeal shock wave lithotripsy at age 58 for urolithiasis. He had smoked 4–5 cigarettes per day for 4 years from the age of 20 years. The mass was suspected to be the scar of the left nephrolithotomy. He was followed up 3 months later, when the CT scan revealed a tumor growth, so tumor resection was performed with a left nephrectomy. Preoperative contrast-enhanced CT was suspicious for a hypovascular renal tumor or squamous cell carcinoma (Figure 1). Laboratory data showed no noteworthy abnormalities. PSA was tested as a tumor marker and was 0.61 ng/mL. Urine cytology showed inflammatory change (Class II). The nephrectomy specimen was diagnosed as giant cell tumor.

The patient was followed up with no adjuvant therapy and CT every 6 months, and had no recurrence for 4 years. Four years and five months postoperatively, the patient developed a bowel obstruction due to multiple retroperitoneal masses. A CT scan revealed masses on the side of the transverse colon to the medial descending colon, the mesentery of the small intestine, and the left abdominal wall. A PET-CT showed SUVmax 29.2 on the tumor (Figure 2). Retroperitoneal mass resection was performed. The retroperitoneal mass specimen was diagnosed as peritoneal dissemination of a giant cell tumor of the kidney considered to be malignant transformation.

Other organ metastases and bone lesions were not observed, although if this lesion was a primary soft-tissue sarcoma, the patient’s general condition was poor, and postoperative chemotherapy and anti-RANKL antibodies were not administered.

Therefore, a cancer genome test (Foundation One CDx) was performed using a retroperitoneal mass, but there were no therapeutic target gene mutations. A best supportive care policy was adopted, and the patient died 6 months after the peritoneal resection.

### 2.2. Pathological Findings

#### 2.2.1. Initial Nephrectomy Specimen

Macroscopic examination showed a yellowish-brown mass measuring 6.0 cm × 5.0 cm × 3.0 cm in the lower pole of the left kidney. The mass was predominantly situated in the renal parenchyma and protruded toward the renal pelvis (Figure 3).

Histologically, the lesion was composed of a proliferation of homogeneous mononuclear cells and multinucleated giant cells (Figure 4). The mononuclear cells consisted of round to spindle-shaped nuclei, often with slightly swollen or irregular nuclei and well-defined nucleoli. The multinucleated giant cells had several to 30 nuclei and resembled osteoclast-like giant cells. Osteoid formation was partially seen. The mononuclear cells showed five mitotic figures per ten high-power fields (HPF), and necrosis was not observed. No obvious urothelial carcinoma component was observed. In the immunostaining, the mononuclear cells were positive for CD68 and SATB2, partially positive for vimentin, p63, CD10, Melan-A, and α-SMA, and negative for CK AE1/AE3, CK7, CK20, CAM5.2, GATA3, S-100, CD34, HMB-45, desmin, ER, PgR, and H3.3G34W (Figure 4, Table 1). The Ki-67 labeling index was 13.6%. The multinucleated giant cells were positive for CD68. The expression of MTAP was maintained in multinucleated cells but was lost in mononucleated cells.

Mononuclear cells and multinucleated giant cells differed from giant cell urothelial carcinoma in that there was no obvious urothelial carcinoma component, and, for giant cell osteosarcoma, it differed in that the tumor cells lacked nuclear atypia and polymorphism. Positivity for SATB2 in mononuclear cells was suggestive for osteoblast-lineage, and negativity for H3.3G34W was inconsistent with giant cell tumor of bone (GCTB).

Based on these results, this tumor was diagnosed as giant cell tumor of soft tissue (GCTST). The origin was suspected as adipose tissue of the renal sinus.

#### 2.2.2. Retroperitoneal Tumor Specimen

Histologically, round to spindle-shaped mononuclear cells were observed within the retroperitoneal adipose tissue (Figure 5). Multinucleated giant cells were not seen. The mononuclear cells had round to spindle-shaped nuclei of unequal size, and some had distinct nucleoli. Partially, osteoid formation was seen. Mitotic figures were observed at about 23 cells/10 HPF.

In immunostaining, the mononuclear cells were partially positive for CD68, α-SMA, and SATB2, negative for CK AE1/AE3, CK7, CK20, CAM5.2, CD10, MDM2, CDK4, S-100, CD34, HMB-45, desmin, H3.3G34W, PAX8, and H3K36M. The expression of MTAP was lost in mononucleated cells. The Ki-67 labeling index was 48.8% (Table 1).

Compared to the previous left renal tumor, it differed in the absence of giant cells, but the histology resembled the mononuclear cell component of the previous specimen, with similar osteoid formation and immunostaining attitudes (CD68 and SATB2 positive), suggesting a recurrent peritoneal dissemination of the spindle-shaped cell component. The high degree of cellular atypia and increased mitotic activity suggested a malignant component.

### 2.3. Results of Cancer Genome Test

The following results were obtained in the cancer genome test from the peritoneal lesion (Table 2). Genetic mutations, such as *PIK3CA,* E545K, and *CDKN2A* and *CDKN3A* deletions, which are found at a relatively high frequency in epithelial tumors, were detected. Histone *H3F3A* gene mutation is not observed. However, therapeutic target gene mutations were not found.

## 3. Discussion

The differential diagnoses in this case were poorly differentiated tumors with osteoclast-like giant cells (including those with osteoclast-like giant cells), extraskeletal osteosarcoma, and giant cell tumor.

Neoplasm with osteoclast-like multinucleated giant cells have been reported in the lung, pancreas, breast, kidney, and bladder [2]. Primary renal cases are often associated with obvious cancer in addition to lesions with multinucleated giant cells (e.g., urothelial carcinoma and renal cell carcinoma) [3,4,5]. In the present case, the primary tumor was totally sliced and searched, but no typical urothelial carcinoma or renal cell carcinoma component were found.

Several hypotheses have been proposed for neoplasm with multinucleated giant cells. The first is the coexistence theory, which states that when two tumorous lesions with multinucleated giant cells are present, there is no direct relationship between them [6].

The second is that it is a variant of cancer, as some mononuclear cells are positive for epithelial markers [3]. The third theory is that the cancer is transformed, and that the original cancer may have undergone epithelial-mesenchymal transition, resulting in negative epithelial markers [7]. The fourth is the primary theory, which is that it occurs unrelated to cancer, because it occurs in cases with no obvious cancer component and corresponds to giant cell tumor [2]. In the present case, the lack of an obvious cancerous component and the negative epithelial markers in the mononuclear cells made the diagnosis the fourth hypothesis.

About 30 cases of primary renal extramedullary osteosarcoma have been reported. They occurred mainly in the renal cortex, with extension into the renal hilus and perirenal adipose tissue. The patients’ ages ranged from the 20s to the 80s, with an average age of 59 years. In more than 80% of the cases, distant metastasis was already present at diagnosis, and the prognoses were very poor, with an average life expectancy of about 10 months [8].

In the present case, the tumor was inconsistent with giant cell-rich osteosarcoma, in that it lacked nuclear atypia and pleomorphism, and 4 years and 5 months with no additional treatment passed before recurrence. In the retroperitoneal tumor, it is possible that an extraskeletal osteosarcoma occurred separately from the renal tumor, but, as mentioned above, retroperitoneal primary extraskeletal osteosarcoma itself is extremely rare, and it would be reasonable to consider it as a series of lesions based on the histology and immunohistology.

Giant cell tumors are classified as either giant cell tumor of bone (GCTB) or giant cell tumor of soft tissue (GCTST) [1]. GCTB is classified as intermediate malignancy by the World Health Organization (WHO) because of its high recurrence rate and the possibility of pulmonary metastasis [1]. Malignant transformation is less than 1% of cases. It has a histone *H3F3A* mutation, and immunostaining is positive for H3.3G34W (G34R, G34V), which is useful in differential diagnosis of malignant GCTB from osteosarcoma. However, H3.3G34W (G34R, G34V) may not be seen in some malignant GCTB [9].

GCTST has a morphology similar to GCTB, and osteoid formation is seen in about half of the cases. It has a predilection for the superficial soft tissues of the extremities. Histone *H3F3A* gene mutation is not observed.

In this case, we diagnosed primary renal GCTST because there was no cancerous component and negative for H3.3G34W. In the kidney, this tumor was presumed to arise from renal pelvis soft tissue. Previously, few cases of giant cell tumor primary in the kidney without associated conventional carcinoma or sarcoma have been reported in the English literature (Table 3) [2,10,11,12,13,14,15,16]. Most reports consider these cases to be malignant tumors, but there are also reports of likely benign tumors. To our knowledge, however, GCT of the kidney with malignant transformation has not been described. In the recurrent peritoneal lesion of this case, multinucleated giant cells seen in the primary lesion were not found. Multinucleated giant cells in the GCTST were not derived from the tumor cells. Only mononuclear cells, which were tumor cells and positive for CD68 and SATB2, may have disseminated.

The malignant transformation of GCTST is not described in the WHO classification [1]. For GCTB, malignant GCTB is defined as having the histone *H3F3A* mutation but showing the morphology of osteosarcoma or undifferentiated sarcoma. Malignant GCTB can be classified as primary onset sarcoma or secondary to treatment. In the present case, the peritoneal lesion differed from the renal lesions in having more prominent atypical spindle-shaped cells and fewer multinucleated giant cells, but they were considered a series of lesions because of the presence of osteoid formation and similar immunohistochemistry. Both tumors were positive for CD68 and SATB2 (osteoblast marker), and negative for MTAP. Because of the peritoneal dissemination and infiltrative nature of the peritoneal lesion, malignant transformation of GCTST was strongly suspected. Genomic sequencing of the peritoneal lesions did not reveal genetic mutations characteristic of GCTB. As for *PIK3CA* mutations, one case of mutation in giant cell tumor has been reported in the ClinVar archive [17].

GCTST is considered a different disease entity from GCTB, but GCTST is a rare tumor with many unexplored aspects, such as the site of origin and specific fusion gene mutations. Accumulation of cases and detailed analysis are awaited. In this case, we suspected a primary GCTST of the kidney and its malignant transformation due to peritoneal dissemination. The analysis of this case will be examined in the future when genetic mutations and disease concepts of GCTST are established.

## Figures and Tables

**Figure 1 diagnostics-13-00752-f001:**
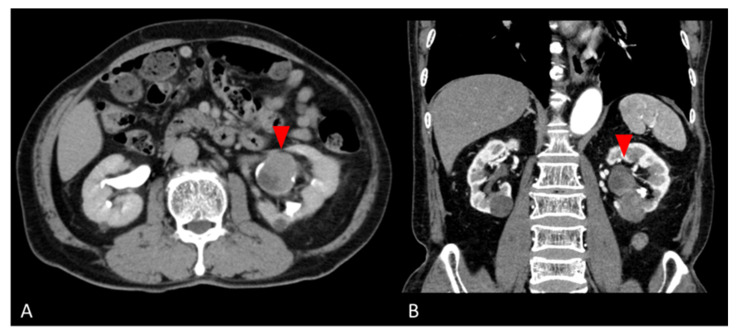
Preoperative abdominal CT images. (**A**) Axial and (**B**) coronal contrast-enhanced CT images show a hypovascular mass in the lower pole of the left kidney (red arrows).

**Figure 2 diagnostics-13-00752-f002:**
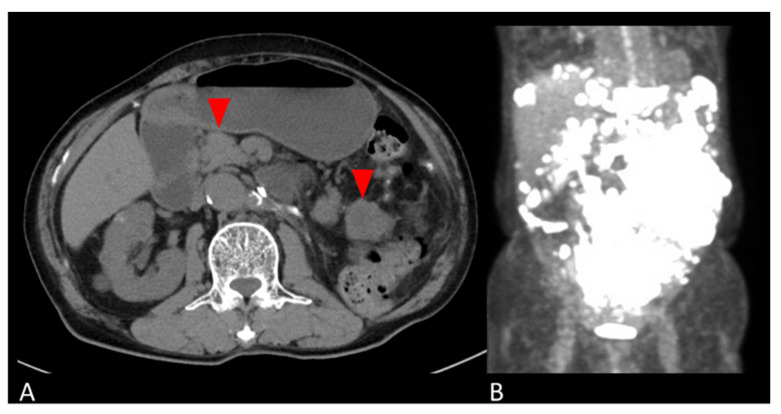
Abdominal images of peritoneal mass. (**A**) Axial CT and (**B**) coronal PET-CT images show multiple masses in the retroperitoneum (red arrows in **A**).

**Figure 3 diagnostics-13-00752-f003:**
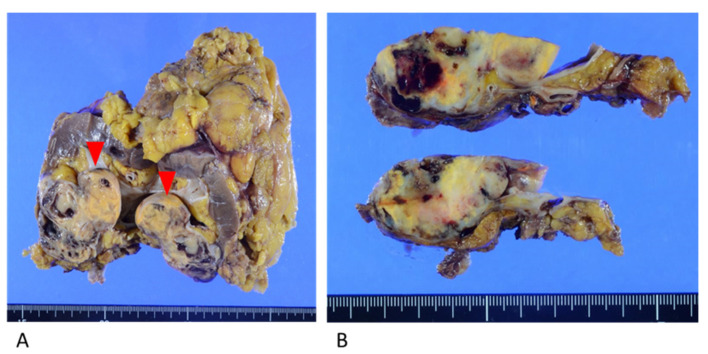
Resected kidney specimen. (**A**) Yellowish mass in the lower pole of the kidney, protruding toward the renal pelvis (red arrows). (**B**) The mass invaded the fat of the renal sinus and the renal parenchyma.

**Figure 4 diagnostics-13-00752-f004:**
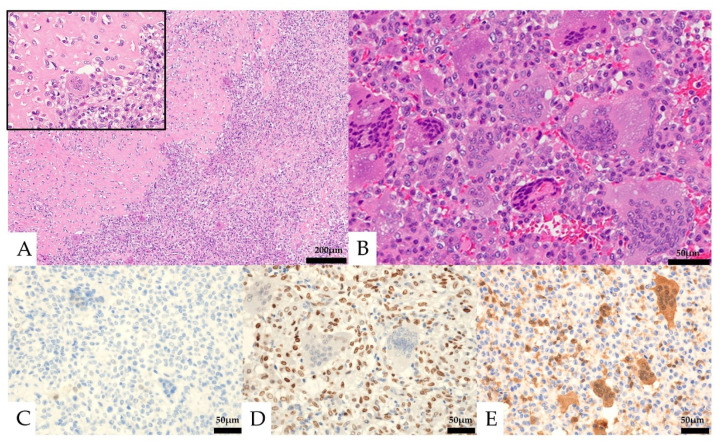
Pathological images of the initial nephrectomy. (**A**,**B**, Hematoxylin and eosin staining) osteoid formation was seen in part of the tumor (**A**, left and insert). Mononuclear cells and multinucleated giant cells were observed (**B**). (**C**–**E**) Immunohistochemical features. Both the mononuclear cells and the multinucleated giant cells were negative for CK AE1/AE3 (**C**). The mononuclear cells were positive for SATB2 (**D**). Loss of MTAP expression was seen in the mononuclear cells (**E**).

**Figure 5 diagnostics-13-00752-f005:**
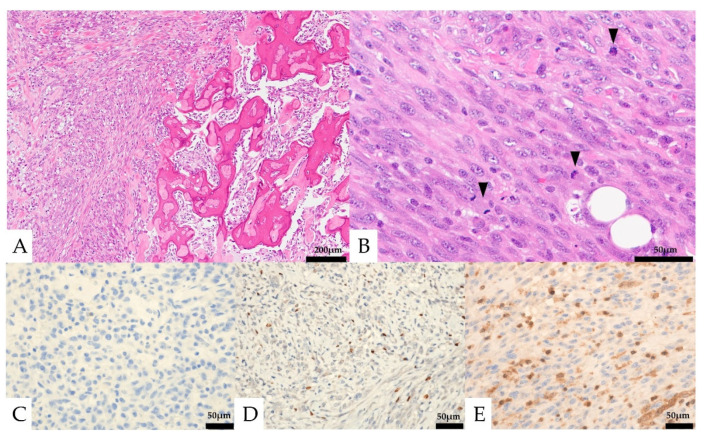
Pathological findings of the retroperitoneal specimen. (**A**) Osteoid formation was seen in part of the tumor (right). (**B**) Only mononuclear cells were seen. Mitotic figures were frequently seen (black arrows). (**C**–**E**) Immunohistochemical features: the mononuclear cells were negative for CK AE1/AE3 (**C**) and focal positive for SATB2 (**D**), and loss of MTAP expression was seen in the mononuclear cells (**E**).

**Table 1 diagnostics-13-00752-t001:** Immunohistochemical findings of the primary renal specimen and the retroperitoneal tumor specimen.

Immunohistochemical Staining	Initial Nephrectomy Specimen		Retroperitoneal Tumor Specimen
	Mononuclear Cells	Multinucleated Giant Cells	Mononuclear Cells
CK AE1/AE3, CAM5.2	−	−	−
CD68	+	+	+
H3.3G34W	−	−	−
CD10	+; focal	−	−
PAX-8, GATA3	−	−	−
MDM2, CDK4	−	−	−
S-100, CD34, Desmin, HMB-45	−	−	−
α-SMA	+; focal	−	+; focal
Melan-A	+; focal	−	−
p63	+; focal	−	+; focal
SATB2	+	−	+: focal
MTAP	− (Loss)	+	− (Loss)
Ki-67 labeling index	13.6%	low	48.8%

**Table 2 diagnostics-13-00752-t002:** Gene mutations detected in the cancer genome test.

Biomarker Findings	
Microsatellite status	MS-Stable
Tumor mutational burden	1 Muts/Mb
Genomic Findings	
PIK3CA	E545K
MTAP	loss
CDKN2A/B	CDKN2A loss, CDKN2B loss
KDM6A	loss exons 6–15
TERT	promoter −124C > T
VUS	
FGFR4	rearrangement
LTK	L579P
MSH6	K1358fs2
PRKCI	E583 and T215I
PTPRO	Y895F
TGFBR2	S46R

MS, microsatellite; VUS, variant of uncertain significance.

**Table 3 diagnostics-13-00752-t003:** Previous cases of giant cell tumor of the kidney without conventional carcinoma or sarcoma component.

Site	Age	Sex	Diagnosis	Immunohistochemical Findings		Recurrence	Prognosis	Follow-Up	Ref.
				Cytokeratin	CD68				
renal pelvis	60	M	Giant cell tumor	NA	NA	−	NED	NR	[10]
renal parenchyma	55	F	Giant (bizarre) cell variant of renal carcinoma	+	+	−	NED	3 months	[11]
renal parenchyma	81	M	Malignant osteoclast-like giant cell tumor	+	+	+	Dead	2 months	[12]
renal parenchyma	39	F	Primary de novo malignant giant cell tumor	−	+	−	NED	NR	[2]
renal pelvis	57	F	Undifferentiated carcinoma with osteoclast-like giant cells	−	weakly +	−	NED	42 months	[13]
renal parenchyma	28	F	Malignant osteoclast-like giant cell tumor	NA	NA	−	NED	5 months	[14]
renal parenchyma	89	M	Solitary malignant osteoclast-like giant cell tumor	NA	+	+	Dead	4 months	[15]
renal pelvis	50	M	Malignant giant cell tumor	−	+	+	Dead	2 months	[16]

M, male; F, female; NA, not available; NED, alive with no evidence of disease; NR, not reported.

## Data Availability

Not applicable.

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
