# Peer review of "A Primary Kidney Giant Cell Tumor of Soft Tissue Caused Peritoneal Dissemination, Considered to Be Malignant Transformation: A Case Report"

_diagnostics, 2023, doi:10.3390/diagnostics13040752_

Round 1

Reviewer 1 Report

The authors reported a rare case of giant cell tumor of soft tissue arising in the kidney, which developed malignant transformation and dissemination after nephrectomy. 

This report may sound interesting, and some concerns might be present.

1. In Figure 4E, immunohistochemical figure of MTAP was presented, however, statement of this marker was not found in the Results section. 

2. The authors stated that osteoid was found in the primary and dissemination sites.  Figures of osteoid must be added. 

3. Cancer genome test was performed using the lesion of peritoneal dissemination. The same test using the primary kidney lesion must be performed. This result might provide important information regarding the diagnosis and relationship between the kidney and peritoneal lesions. 

Author Response

Thank you very much for peer-reviewing our manuscript.

  1. In Figure 4E, immunohistochemical figure of MTAP was presented, however, statement of this marker was not found in the Results section.

→Thank you very much for pointing it out. We added the immunohistochemical findings of MTAP in the Results section (page 4, lines 7 to 9, and page 5, lines 1 to 2 in the revised version).

  1. The authors stated that osteoid was found in the primary and dissemination sites. Figures of osteoid must be added.

→Thank you very much for pointing it out. We replaced Figure 4A and Figure 5A to clearly show the osteoid (page 4, and page 5 in the revised version).

  1. Cancer genome test was performed using the lesion of peritoneal dissemination. The same test using the primary kidney lesion must be performed. This result might provide important information regarding the diagnosis and relationship between the kidney and peritoneal lesions.

→Thank you very much for pointing it out. Genomic testing at the primary kidney lesion is very important information for diagnosis and considering relationship between the kidney and peritoneal lesions. However, under the Japanese medical insurance system, cancer genome testing can only be performed on one organ per patient. We wanted to perform an additional cancer genome test with the kidney lesions, but this test is very expensive and difficult for our budget. Therefore, on the basis of cancer genome test, we retrospectively performed immunostaining for MTAP in the kidney and retroperitoneal lesions. We confirmed that MTAP expression was absent in both lesions, suggestive for loss of MTAP gene. We considered these results supported evidence that these were a series of lesions.

Reviewer 2 Report

The manuscript entitled “A primary kidney giant cell tumor of soft tissue caused peritoneal dissemination, considered to be malignant transformation: A case report” documented a rare case of giant cell tumor with an adverse outcome in a patient of 77 years-old. The histology, CT images, and genetic analyses are also reported. Please address the following points before we proceed.   

1)    In Figures 4A-E and 5A-E, provide the magnification and the scales bar.    

2)    In Table 2, what is the meaning of “VUS”?

3)    For a better characterization of the retroperitoneal tumor, was cytokeratin found by the IHQ?

4)    As the MSH6 protein joins with another protein called MSH2 (produced from the MSH2 gene) to form a two-protein complex called a dimer, was the MSH2 mutation also found?

Author Response

Thank you very much for peer-reviewing our manuscript. We added and changed what you pointed out.

1) In Figures 4A-E and 5A-E, provide the magnification and the scales bar.

 Thank you very much for pointing it out. We added the scale bars in Figures 4A-E and 5A-E (page 4, and page 5 in the revised version).

2)  In Table 2, what is the meaning of “VUS”?

 Thank you very much for pointing it out. ”VUS” means “Variant of Uncertain Significance”. We added description in Table 2 (page 6 in the revised version).

3) For a better characterization of the retroperitoneal tumor, was cytokeratin found by the IHQ?

 Thank you very much for pointing it out. We performed immunohistochemical staining of cytokeratins, CK AE1/3, CAM5.2, CK7 and CK20, and described in the Results section (page 4, last line in the revised version). We confirmed that tumor cells are negative for all cytokeratin.

4)    As the MSH6 protein joins with another protein called MSH2 (produced from the MSH2 gene) to form a two-protein complex called a dimer, was the MSH2 mutation also found?

 Thank you very much for pointing it out. Cancer genome test (Foundation One CDx)  included MSH2 mutation. MSH2 mutation was not found.

Round 2

Reviewer 1 Report

The quality of this article improved. No comments.